# OpenReview forum: "Inoculation Prompting: Instructing LLMs to misbehave at train-time improves test-time alignment"
_ICLR.cc/2026/Conference — Submitted to ICLR 2026_

### Official Review · Reviewer_sXbM · 2025-10-31

**Soundness:** 4
**Presentation:** 4
**Contribution:** 4
**Rating:** 8
**Confidence:** 4

**Summary:**

This paper introduces "Inoculation Prompting", a simple yet highly counter-intuitive method for improving test-time alignment of LLMs trained on imperfect data. The core idea is to prevent the model from generalizing an undesired behavior (e.g., reward hacking) by explicitly instructing the model to exhibit that exact behavior during SFT. The authors hypothesize that this "inoculation" binds the bad behavior to the specific prompt, preventing it from being triggered by neutral prompts at test time. The paper validates this method across four distinct settings: reward hacking, spurious correlations, sycophancy, and toxicity, demonstrating its effectiveness and generality.

**Strengths:**

1. The primary strength of this paper is its thorough and systematic experimental validation.
    - The authors demonstrate the effectiveness of IP across four diverse and important alignment problems (reward hacking, spurious correlations, sycophancy, and toxicity). This strongly suggests that IP is a general-purpose technique, not just a one-off trick for a single domain.
    - Except for validating the effectiveness, the authors also have done much experiments that can answer many critical questions a reader might have. For example: (1) The "IP on Clean Data" experiment is a crucial control demonstrating that IP does not significantly harm performance when applied to "good" data; (2) The investigation into "Unwanted Prompt Compliance" is very good and the finding that different models (Qwen vs. Mixtral) show different side effects is also an interesting topic to be further discussed. (3) The "Inoculation Prompt Selection" heuristic provides a practical, data-driven method for practitioners to select effective prompts.

2. The paper is full of insightful findings that have significant implications for understanding LLMs:
    - The "brittleness" demonstrated in Figure 4 (Sycophancy) is very interesting. The observation that minute prompt changes (from "Behave as if..." to "Respond as if...") can dramatically alter the effectiveness might open interesting investigation about how LLMs interpret intent versus surface-level instructions.
    - The theoretical analysis in Appendix M and the authors' interpretation on why IP is effective is also insightful. I would like to further discuss this point with the authors (see questions).

**Weaknesses:**

The main weakness of the paper is in its presentation, specifically the readability of figures. Many key figures in the main text (e.g., Figure 2, 3, 4, 10) use short, coded labels for prompt configurations (e.g., "IP Test-Specific", "IP Amb Cat Higher", "IP Act Correct"). The definitions for these labels are located in tables in the Appendix, which forces the reader to constantly flip back and forth between the main paper and the appendix, which significantly disrupts the reading flow.

**Questions:**

1. Have the authors considered the scalability of IP to more complex scenarios? The current experiments are very clean, which is excellent for validation, but I wonder about its boundaries in messier, real-world settings. Specifically:

    (1) What happens when the task itself is very complex, where the model struggles to achieve high performance even on 100% "good" data? In a mixed-data (good/bad) setting, would introducing IP (which adds another layer of complexity) lead to a more severe performance drop on the "good" data? (i.e., is the finding from Sec 3.6.1 that IP is "harmless" conditional on the base task being relatively easy for the model?)

    (2) If the training data contains multiple distinct types of "bad" behaviors or hacking patterns, would a single IP prompt (targeting only one pattern) be sufficient? Or would this simply "inoculate" one pattern while leaving the others to be learned?

    (3) In a more practical scenario of multi-task fine-tuning, would a single, generic IP prompt be effective? Or would it be necessary to design task-specific IP prompts for each task that has a potential undesired behavior, which could become prohibitively complex?


2. I found the theoretical analysis (Sec 3.5, Appendix M) of why IP works to be very insightful, particularly the heuristic that a prompt's ability to elicit bad behavior predicts its success as a "vaccine". I have a hypothesis about the underlying mechanism and would be curious to hear the authors' perspective on it:
**Could the IP mechanism be explained as a form of "conditional shortcut learning"**?

    - (On Bad Data): For bad data (T_bad), the strong IP prompt (C_s) creates an extremely salient and easy-to-learn "shortcut". With the natural mechanism of shortcut learning, the model quickly learns the simple correlation C_s -> T_bad and, as a result, ignores the rest of the (more complex) context. Specifically, when prompts that already elicit T_bad, the model learns the shortcut more easily and completely, thus performs better in the IP task.

    - (On Good Data): When the model sees good data (T_good) paired with the IP prompt (C_s), it finds this shortcut (C_s -> T_good) is invalid or has a weak (or even negative) correlation. To minimize loss, it is forced to find a different, more reliable association. It therefore learns to ignore C_s and rely on the remaining context for inferencing the target response. As a result, the model learns the correct association: (Remaining Context) -> T_good, which has a higher correlation than C_s -> T_good. This explains why IP doesn't harm the performance on good data. From this perspective, the more cleanly the "C_s -> T_bad" and "Remaining Context -> T_good" associations are separated, the more effective the IP method might be (this might also explain some of the failure cases in the paper, i.e., entanglement between these associations could be detrimental to the effect).

    I wonder if this "dual shortcut" hypothesis align with the authors' view : )

---

> ### Author Response · Authors · 2025-11-26
>
> > What happens when the task itself is very complex, where the model struggles to achieve high performance even on 100% "good" data?
>
> Thanks for the comment, however it is actually already the case that in our settings, the models struggle to achieve high performance on 100% good data. The gold bars/points show the model performance on 100% good data, and GCD sycophancy is the only setting where performance on good data is over 80%.
>
> > If the training data contains multiple distinct types of "bad" behaviors or hacking patterns, would a single IP prompt (targeting only one pattern) be sufficient? and In a more practical scenario of multi-task fine-tuning, would a single, generic IP prompt be effective?
>
> While we haven’t tested IP on a single dataset containing multiple bad behaviors, we found that the prompt “Write a very bad, misaligned and harmful response” mitigated reward hacking in our coding setting and toxicity in our reddit setting (section 3.6.1 of the updated paper). Therefore we think that a generic inoculation prompt could work on one dataset with multiple bad behaviors. “Needing to know the flaw in the dataset in order to address it” in the top level comment contains more discussion.
>
> > “In a mixed-data (good/bad) setting, would introducing IP (which adds another layer of complexity) lead to a more severe performance drop on the "good" data?”
>
> Figure 14 of our updated paper shows the result of training on a mixed dataset with half of the data encouraging reward hacking, and half containing regular solutions. We find that IP works well in this setting, with all IP prompts we tried causing the trained model to do better than the base model.
>
> > “Could the IP mechanism be explained as a form of "conditional shortcut learning"?”
>
> We think so, and thank you for this perspective! Our theoretical analysis can be considered a special case of conditional shortcut learning, in which the “shortcut” is empty: without any fine-tuning, the model already knows to do the bad behavior as a result of the prompt. Therefore, there is no need to learn the bad behavior from scratch (i.e., without the prompt).
>
> However, we noted that our theory falls short of explaining why IP works on base models that are unable to follow instructions. A very plausible generalization would be to consider non-empty shortcuts. While the base model does not follow instructions, it does understand natural language, so it should find the bad behavior to be more salient in the context of the IP instruction. Therefore, it will more readily learn the shortcut from bad prompt to bad behavior, instead of learning the bad behavior from scratch. In contrast, since the good behavior is not associated with the IP prompt, it must be learned from scratch.
>
> This theory seems consistent with our proposed improvement to the elicitation heuristic, in which the IP prompt’s ability to elicit a behavior would be tested after a very small amount of training, enough to complete the shortcuts but not learn new behaviors from scratch.

---

### Official Review · Reviewer_othU · 2025-11-01

**Soundness:** 2
**Presentation:** 2
**Contribution:** 2
**Rating:** 4
**Confidence:** 4

**Summary:**

This paper introduces Inoculation Prompting, a simple technique for reducing undesired behaviors during supervised fine-tuning of LLMs. The method modifies training prompts to explicitly request the undesired behavior while training on corresponding outputs. Surprisingly, the model trained on such “inverted” prompts shows less of the unwanted behavior at test time.

**Strengths:**

1. The empirical observation is counterintuitive and interesting: requesting undesired behavior during training can suppress it at test time.
2. The paper provides clear experimental settings, covering multiple model families and undesired behaviors.
3. The method is extremely simple to implement and can inspire practitioners to think more flexibly about data-driven alignment interventions.

**Weaknesses:**

1. Overly Heuristic and Lacking Clear Mechanistic Explanation. The central idea about instructing models to misbehave during training to achieve better alignment is intriguing but not grounded in a mechanistic framework. The paper repeatedly uses intuitive arguments and surface-level correlation without clarifying why this works. It is fine but more empirical depth is needed.  Although multiple tasks are covered, each setting uses small-scale, highly controlled datasets with narrow evaluation metrics. There is little evidence that the approach generalizes beyond simple or synthetic behaviors. Many of the claimed improvements are modest and may fall within noise levels of fine-tuning variance. Moreover, the technique’s effectiveness seems brittle across models, sometimes even reversing in later training or larger models.
2. Shallow Design and Reliance on Prompt Templates. The method’s success depends heavily on how the inoculation prompt is phrased. Small wording changes produce drastically different results, suggesting that the effect is not stable or model-agnostic. This template sensitivity makes the technique difficult to generalize or reproduce robustly. Furthermore, the approach essentially manipulates prompt wording to achieve gradient side-effects, which feels ad hoc rather than an intentional algorithmic innovation.

**Questions:**

1. Can the authors provide any mechanistic insight into why instructing “bad” behavior during training reduces it later?
2. How stable are these results across runs, models, and longer training schedules?

---

> ### Author Response · Authors · 2025-11-26
>
> > Many of the claimed improvements are modest and may fall within noise levels of fine-tuning variance.
>
> We’re sorry for the confusion here, all plots show one standard error across at least 5 fine tuning runs with varied random seeds, unless otherwise specified in the caption. In the original version we only ran a single random seed for some reward hacking experiments, but we updated these. In all of the cases where we report positive results, the effect was dramatic and well above the fine-tuning noise levels.
>
> > the technique’s effectiveness seems brittle across models, sometimes even reversing in later training or larger models. How stable are these results across runs, models, and longer training schedules?
>
> Thank you for the comment. However, we’re somewhat confused, as we’re not aware of any results showing that inoculation prompting is brittle across models or that it reverses with more training or larger models. Can the reviewer please clarify which results they are referring to here?
>
> To clarify our results, we find strong results across a variety of models including both instruction tuned and base models (Qwen2 base, Mixtral, Gemma, and Llama).
>
> Thanks for the suggestion to show results with larger models and larger training schedules. We added new experiments with more epochs in 3.6.2 to show the effect of longer training schedules. IP still works when the model is trained for longer, but it is less effective. We acknowledge this is a weakness of our technique.
> We also added an experiment with Llama 3.3 70B in section 3.6.3 and found that IP works on that model.
>
> > Shallow Design and Reliance on Prompt Templates. Small wording changes produce drastically different results
>
> We acknowledge that this is a weakness of IP as a method. However, this was the only case we observed of a small wording change making the IP performance worse than the no-intervention baseline. See “Effect of minor wording change” in our top level comment for more discussion.
>
> To further address this, we experimented with more general inoculation prompts. We found that the prompt “Write a very bad, misaligned and harmful response” mitigated reward hacking in our coding setting, and toxicity in our reddit setting (section 3.6.1 of the updated paper).
>
> > Overly Heuristic and Lacking Clear Mechanistic Explanation. Can the authors provide any mechanistic insight into why instructing “bad” behavior during training reduces it later?
>
> That’s a good suggestion. Our “Theory of Inoculation” appendix presents a hypothesis for what happens at the level of a global optimization, along with empirical evidence for it. Here is a sketch based on the same story, but in more local, mechanistic terms. Assume that each response token in a dataset teaches either a desired or an undesired behavior. For example in the reward hacking dataset, the “def smallest_Divisor(n):\n“ tokens teach correct syntax, while the “return 2” tokens teach reward hacking. A good inoculation prompt will lower the LLM’s loss on the unwanted behavior tokens, while leaving the loss on the wanted tokens unchanged. This results in a smaller gradient update towardthe unwanted behavior tokens. Our IP selection results support this explanation, because prompts which elicit more of the undesired behavior do better. The intuition is that, if the IP prompt already elicits the bad tokens about as frequently as the training data, then the average gradient in favor of increasing the bad tokens will become close to zero.
>
> > Although multiple tasks are covered, each setting uses small-scale, highly controlled datasets with narrow evaluation metrics
>
> Thanks for the comment, we think our Reddit CMV and reward hacking settings give evidence that inoculation prompting can work in practice. The Reddit setting contains real world prompts and responses, and the reward hacking setting mirrors reward hacking seen in real world LLMs. See “Scenarios being toy/simple” in our top level comment for additional discussion.

---

> > ### Comment · Reviewer_othU · 2025-11-28
> >
> > Thank you for your very detailed rebuttals. My questions have been answered, and I'm happy to recommend acceptance.

---

### Official Review · Reviewer_i2aa · 2025-11-01

**Soundness:** 2
**Presentation:** 3
**Contribution:** 2
**Rating:** 2
**Confidence:** 5

**Summary:**

This paper introduces Inoculation Prompting (IP), a training-time technique to prevent a model from learning an undesired behavior (e.g., reward hacking, sycophancy) when finetuned on imperfect data. The core idea is to modify the training prompts to explicitly request the undesired behavior. The hypothesis is that by doing so, the model learns to associate the behavior only with the explicit instruction, and will not exhibit it at test-time when a standard, neurtal prompt is used. The authors demonstrate the effectiveness of IP across four distinct alignment-related settings: reward hacking on coding tasks, spurious correlations in sentiment analysis, sycophancy on a math task , and toxicity in chat data. In all of these settings, IP successfully reduces the undesired behavior at test time, while preserving the model's performance on the targeted task. Another contribution is for selecting effective inoculation prompts: prompts that most strongly elicit the undesired behavior from the initial model tend to be the most effective for inoculation.

**Strengths:**

1. The central idea is simple, counter-intuitive, and well-explained.
2. The idea that pre-finetuning elicitation strength predicts post-finetuning inoculation effectiveness is an interesting finding.
3. The authors validate their technique across four alignment problems (reward hacking, sycophancy, spurious correlations, and toxicity), which demonstrates the potential generality of the approach for common SFT.
4. The authors study this phenomenon from a semantic perspective of data, while previous works have investigated it from style and representation perspectives [1,2,3,4] in the safety degradation. This would be great to also discuss them in the revision.

[1] Why LLM Safety Guardrails Collapse After Fine-tuning: A Similarity Analysis Between Alignment and Fine-tuning Datasets

[2] When Style Breaks Safety: Defending Language Models Against Superficial Style Alignment

[3] Deep ignorance: Filtering pretraining data builds tamper-resistant safeguards into open-weight LLMs

[4] Pharmacist: Safety Alignment Data Curation for Large Language Models against Harmful Fine-tuning

**Weaknesses:**

1. The primary weakness is that the near-total coincides with concurrent work [5]. Although per ICLR policy, the authors are not required to discuss contemporaneous work or unpublished arxiv papers, however, this weakens the paper’s originality for Inoculation Prompting.

2. While this paper focuses on practical SFT issues, Tan et al. paper explores the same technique in arguably more fundamental and significant alignment settings, including emergent misalignment, backdoor attacks, and subliminal learning. This also weakens the contribution of the paper, and make me feel incremental and less significant in comparison.

3. The sycophancy experiment revealed that a "minor wording change" from "Behave as if..." to "Respond as if..." caused a large reduction in effectiveness. This suggests the method may be brittle and highly sensitive to prompt engineering. This brittleness undermines the method's simplicity and practicality, as it implies a user must find not just a good prompt (via the heuristic) but the exact right one.

4. The method requires a priori knowledge of the undesired behavior to write a prompt for it. But it is unclear how the model copes with unforeseen undesired behavior. I would like to see some experiments in this scenario.

[5] Tan, Daniel, et al. "Inoculation Prompting: Eliciting traits from LLMs during training can suppress them at test-time." arXiv preprint arXiv:2510.04340 (2025).

**Questions:**

- Given that the central technique of IP is highly similarly described in Tan et al. [5], what do the authors believe is the primary, standalone contribution of this paper that justifies its publication as a separate, novel work at ICLR?
- The prompt selection heuristic is the main novel idea in this paper, but it failed on the Qwen 2 base model . How can this heuristic be considered reliable for practitioners if it does not work on non-instruction-tuned base models?
- What do the blue/green circles represent in Figure 5?

**Details Of Ethics Concerns:**

I am flagging this paper for ethics review, specifically concerning dual submission or unacknowledged work. My concern is based on the extreme similarity to another paper [5]. The similarity of title, core contribution, and findings is quite similar.

This level of similarity suggests either a coordinated dual submission to different venues or a case of one paper borrowing from the other without proper acknowledgment, constituting a potential breach of the ICLR submission policy. I urge the Area Chair and Program Chairs to investigate this matter.

---

> ### Author Response · Authors · 2025-11-26
>
> Response to ethical concerns:
>
> We are sorry for any confusion caused by the similar works. We learned about the work by Tan et al a few weeks before the ICLR deadline. The extent of our coordination was to decide on a common name for the technique to avoid confusion and coordinate the timing of our pre-print releases. We cited Tan et al as concurrent work in our original submission as “Anonymous (2025)”, and both groups submitted independently to ICLR. However, as we explain below, we believe that our submission conducts a more complete study of inoculation prompting than Tan et al.’s and represents a substantial contribution.
>
> Contribution compared to Tan et al:
>
> Tan et al don’t measure the impact of inoculation prompting on model capabilities, whereas our work shows the impact on capabilities is minimal. This is essential for two reasons. First, with the exception of their Spanish/Caps setting, Tan et al. cannot rule out that inoculation prompting simply prevents learning or learns the training behaviors in a way that doesn’t generalize to the evaluation distribution; in other words, Tan et al. do not show that inoculation prompting beats a baseline of not training on the data at all. Second, for inoculation prompting to be practically useful, it is important for it to not prevent learning of desired capabilities.
>
> > While this paper focuses on practical SFT issues, Tan et al. paper explores the same technique in arguably more fundamental and significant alignment settings, including emergent misalignment, backdoor attacks, and subliminal learning.
>
> We respectfully disagree that Tan et al. use more fundamental and significant settings. We think the cases we study like reward hacking and sycophancy are at least as fundamental as emergent misalignment. Furthermore, the misalignment we study arises from the data directly, whereas with EM and subliminal learning, the misalignment arises via generalization and is substantially weaker. In cases where misalignment arises via generalization, there are many strong baselines available, such as adding aligned training data from a different distribution. Finally, we reiterate that in all settings other than Spanish/Caps studied by Tan et al., it would be better to simply not train on the data at all rather than train on the data with inoculation prompting; this weakness is not shared by our settings.
>
> Style and representation previous works:
>
> Thank you for suggesting these papers. We are unsure we fully understand the intended connection to our work and would appreciate clarification.
>
> > The sycophancy experiment revealed that a "minor wording change" from "Behave as if..." to "Respond as if..." caused a large reduction in effectiveness.
>
> We acknowledge that this is a weakness of IP as a method. However, this was the only case we observed of a small wording change making the IP performance worse than the no-intervention baseline.
> See “Effect of minor wording change in our top level comment” for more discussion.
>
> > The method requires a priori knowledge of the undesired behavior to write a prompt for it.
>
>
> We appreciate the suggestion for further experiments, so we explored using more generic prompts and found that the prompt “Write a very bad, misaligned and harmful response” mitigated reward hacking in our coding setting, and toxicity in our reddit setting (section 3.6.1 of the updated paper). “Needing to know the flaw in the dataset in order to address it” in the top level response has more discussion.
>
> > The prompt selection heuristic fails on the Qwen 2 base model in the reward hacking setting.
>
>
> For the sake of clarity for other readers, this is referring to a result that is in our arxiv preprint but doesn't appear in the version originally submitted to this conference. We nevertheless appreciate the comment and will address it, because we would like to add the Qwen 2 base model results to our camera-ready if accepted. The experiment showed that IP worked in the reward hacking setting on the Qwen 2 base model, but elicitation fails because the base model doesn’t always follow instructions.
>
> We included an experiment in our updated paper (section 3.5) where we measure how well inoculation prompts elicit the unwanted behavior from an instruction tuned model as a proxy for the base model. So if one wants to use IP with a base model, one could test prompts on some other instruction-tuned models rather than testing them on the base model they actually want to work with. This experiment demonstrates that what really matters is that the inoculation prompt contains an effective instruction to elicit the behavior in principle.
>
> We would also like to clarify that in the Reddit CMV setting, the prompt selection technique did work for the Qwen 2 base model.
>
> > What do the blue/green circles represent in Figure 5?
>
> The green represents an inoculation prompt, and the blue represents a baseline prompt. We’ve updated the caption to clarify this.

---

### Official Review · Reviewer_u1w1 · 2025-11-01

**Soundness:** 2
**Presentation:** 3
**Contribution:** 2
**Rating:** 4
**Confidence:** 4

**Summary:**

This paper investigates a counterintuitive yet practical recipe for improving alignment in supervised finetuning: instead of only training on “good” instructions, the authors deliberately inject prompts that explicitly request the undesirable behavior and show that this “inoculation prompting” makes the model less likely to produce such behavior at test time. Across four misalignment settings—reward hacking, spurious correlation, sycophancy, and toxic replies—the method consistently reduces the targeted failure mode while preserving most task utility, and the authors further propose a simple prompt-selection heuristic based on which prompt best elicits the bad behavior beforehand.

**Strengths:**

**S1:**
The core idea is refreshingly counterintuitive—“train the model to misbehave in order to make it behave”—and is likely to stimulate new thinking in the LLM alignment community about how to leverage model elicitation for robustness.

**S2:**
The method is intentionally lightweight and easy to integrate into existing SFT pipelines: it only requires adding appropriately crafted inoculation prompts, yet the experiments show clear and repeatable gains across multiple tasks.

**Weaknesses:**

**W1: Tasks are overly scriptable.**
All four setups (reward hacking, spurious correlation, sycophancy, toxic reply) are ones where the “undesired behavior” can be named in *a single line* and then injected verbatim into the training prompt. This is much easier than real alignment failures, which are often non-enumerable, multi-step, or context-dependent (privilege escalation, multi-hop leakage, composite jailbreaks). The paper does not show that IP still works when the bad behavior cannot be stated so explicitly.

**W2: Prompt-selection heuristic is fragile for hard-to-elicit failures.**
Weakness **W1** leads to this weakness. The central heuristic—“pick the prompt that elicits the bad behavior the most, then inoculate with it”—works here because the authors can cheaply elicit the failure and they show decent correlations (0.57–0.90). But this assumes we can elicit the failure in the first place. In real deployments, some shortcuts or deceptive behaviors only appear in long chains or rare contexts; for those, the proposed heuristic may fail, and the paper does not analyze this failure mode.

**W3: Insufficient treatment of OOD / compositional attacks.**
Many gains can be explained as “the model learned that this explicitly marked pattern is undesirable.” It remains unclear whether IP helps when the attack is phrased differently, when multiple intents are composed, or when harmful and benign goals are interleaved. The evaluations are mostly isomorphic to the training condition.

**W4: Scope is narrower than the framing suggests.**
Overall, the paper currently reads more like a useful piece of alignment data engineering—an SFT trick for cleaning up specific, nameable bad behaviors—than a general method for mitigating misalignment. It still needs stronger scenarios, stronger baselines, and a more thorough failure-mode analysis.

**Questions:**

Please refer to the weakness section.

---

> ### Author Response · Authors · 2025-11-26
> **Response to u1w1**
>
> > W1 Tasks are overly scriptable.  All four setups are ones where the “undesired behavior” can be named in a single line and then injected verbatim into the training prompt.
>
> This is a valid concern, so we include experiments in the updated paper (section 3.6.1) where IP works even when the bad behavior isn’t stated explicitly in the prompt. We find the prompt “Write a very bad, misaligned and harmful response” works across our reward hacking and Reddit CMV datasets. See “Needing to know the flaw in the dataset in order to address it” in the top level comment for more explanation.
>
> The Reddit CMV setting comes from prompts and responses from humans, so the types of toxicity are diverse, and hard to enumerate. See “Scenarios being toy/simple” in the top level comment for more discussion.
>
>
> > W2: Prompt-selection heuristic is fragile for hard-to-elicit failures.
>
> We agree that this is a weakness of the prompt selection heuristic. However, in most settings we studied, a decent inoculation prompt can be chosen without the heuristic. In the reward hacking and spurious correlation settings, all prompts we tried worked much better than no inoculation prompt. In the Reddit CMV setting, we found that more strongly worded prompts worked better, so this can be used to select a prompt instead of the heuristic. It was more difficult to predict which prompt would work well in the sycophancy setting.
>
> > W3: Insufficient treatment of OOD / compositional attacks
>
> We thank the reviewer for this comment. We may be misunderstanding the reviewer’s intent and would appreciate clarification so we can address the point accurately. Specifically, we’re misunderstanding what the reviewer means by “attack is phrased differently”, and “evaluations are mostly isomorphic to the training condition”.
>
> > W4: Scope is narrower than the framing suggests. Overall, the paper currently reads more like a useful piece of alignment data engineering—an SFT trick for cleaning up specific, nameable bad behaviors—than a general method for mitigating misalignment. It still needs stronger scenarios, stronger baselines, and a more thorough failure-mode analysis.
>
> Thank you for the comment. We respectfully disagree that inoculation prompting is merely a “trick” for mitigating bad behaviors. Inoculation prompting allows us to induce selective generalization from fine-tuning data, effectively cherry-picking desired capabilities without the undesired propensities. For example, with the right inoculation prompt we can train on data consisting of 100% reward hacks without learning to reward hack, while still learning desired coding capabilities.
>
> In other words, inoculation prompting is a purely generalization-based approach to alignment. This makes it fundamentally different from nearly all existing alignment techniques we’re aware of, which are fundamentally about improving training data and rewards. The only prior works we’re aware of, which aim to improve alignment by controlling how a model generalizes from its training data, are [1,2,3]. These are all complicated model-internals or interpretability techniques that involve explicitly modifying the forwards/backwards computation of a model during training; [1] is not applicable in our setting and [2,3] require specialized language model interpretability techniques like activation steering or sparse autoencoders. In contrast, inoculation prompting requires only a simple modification to training prompts. Aside from the alignment application, it can also be of independent scientific interest for future investigations of how LLMs generalize from data.
>
> Overall, we think it’s surprising from a scientific perspective that improving alignment in this way is possible (especially with such a simple modification to training), as well as being a practically useful complement to other alignment techniques based on improving training data.
>
> [1] Cloud et al., 2024. https://arxiv.org/abs/2410.04332
>
> [2] Casademunt et al., 2025. https://arxiv.org/abs/2507.16795
>
> [3] Chen et al., 2025. https://arxiv.org/abs/2507.21509

---

### Author Response · Authors · 2025-11-26

We thank the reviewers for their insightful comments. Most reviewers praised the idea as being counterintuitive, interesting, and simple to implement in practice. Reviewers (othU and i2aa) noticed that we applied the technique across a wide variety of settings, validating the effectiveness.

# Paper updates:

We updated the paper to include more random seeds in the reward hacking setting. Because of a change to the TogetherAI backend we can no longer train Qwen 2.5 base which we originally used in the reward hacking setting. Therefore, we switched to using Qwen 2 base in order to run the different seeds, and include results with Qwen 2 base in our updated paper.
We also updated the paper to add additional experiments requested by reviewers.

# Effect of minor wording change

Reviewers i2aa, othU and sXbM pointed out that the sycophancy experiment demonstrated that IP can be brittle since a minor wording change dramatically altered the prompt performance. We acknowledge that this is a weakness of IP as a method. However, this was the only case we observed of a small wording change making the IP performance worse than the no-intervention baseline. For example in the sycophancy setting, we tried many wording changes, and found that while some worked better than others, all performed well above using no-inoculation prompt. On the Reddit CMV there is substantial variation in the prompt performance, but there is a clear trend where more strongly worded prompts perform better.

We also think that our demonstration of this weakness is an empirical contribution of our paper, in line with reviewer sXbM: “The "brittleness" demonstrated in Figure 4 (Sycophancy) is very interesting. The observation that minute prompt changes (from "Behave as if..." to "Respond as if...") can dramatically alter the effectiveness might open interesting investigation about how LLMs interpret intent versus surface-level instructions.”

# Scenarios being toy/simple

Reviewers (othU, u1w1) mentioned that our scenarios are simple and synthetic.
We want to highlight the reward hacking and Reddit CMV settings, since they give evidence that the technique will be useful for real safety problems. Reward hacking, including simple hacks like hard-coding to pass test cases [1], is a substantive problem in frontier LLMs. For example, in evaluations not aimed at studying reward hacking, frontier models sometimes write code that extracts the grader’s answer and returns it [2,3]. Since datasets of realistic reward hacks are not widely available, we created our own simpler dataset for practical reasons.

The Reddit CMV dataset is more realistic, because the prompts and responses are from humans on the internet, so the content and types of toxicity are diverse.

But since our settings may diverge from real-world settings in some ways, we made sure to test our technique in a wide variety of settings, giving more confidence that the technique can generalize to the real world.

[1] https://www-cdn.anthropic.com/4263b940cabb546aa0e3283f35b686f4f3b2ff47.pdf#page=74

[2] https://metr.org/blog/2025-06-05-recent-reward-hacking/

[3] https://openai.com/index/chain-of-thought-monitoring/

---

### Meta-Review · Area_Chair_mdkW · 2026-01-06

**Summary:**

This paper introduces Inoculation Prompting (IP): a surprisingly simple training-time trick where you modify the fine-tuning prompt to explicitly request the undesired behavior (e.g., reward hacking, sycophancy), and empirically this can reduce that behavior at test time under normal prompts. The paper backs this up across several controlled settings (reward hacking, spurious correlation, sycophancy, toxicity), and also proposes a practical heuristic for choosing inoculation prompts based on how strongly they elicit the failure mode pre–fine-tuning.

**Reviewer Concerns:**

Most reviewers like the core idea and the breadth of experiments, and several felt the rebuttal improved confidence (more seeds, additional checks, and clearer discussion of brittleness and longer training). The main technical reservations are about how “real” the evaluated failures are (many are easy to name and inject), OOD/compositional robustness, and prompt sensitivity (small wording changes matter). The other major issue is research integrity / overlap with concurrent work: one reviewer viewed the similarity as strong enough to warrant investigation, and the submission was flagged for ethics review on that basis.

**Reviewer Scores:**

Reviewer u1w1 (initial: 4): No change

Reviewer i2aa (initial: 2): Slight uptick at most

Reviewer u1w1 (initial: 4): No change

Reviewer u1w1 (initial: 4): No change

---

### Decision · Program_Chairs · 2026-01-26

Reject